# Biological Activity of c-Peptide in Microvascular Complications of Type 1 Diabetes—Time for Translational Studies or Back to the Basics?

**DOI:** 10.3390/ijms21249723

**Published:** 2020-12-20

**Authors:** Aleksandra Ryk, Aleksandra Łosiewicz, Arkadiusz Michalak, Wojciech Fendler

**Affiliations:** 1Department of Biostatistics and Translational Medicine, Medical University of Lodz, 92-215 Lodz, Poland; aleksandra.ryk1@stud.umed.lodz.pl (A.R.); losiewiczaleksandra@gmail.com (A.Ł.); arkadiusz.michalak.lek@gmail.com (A.M.); 2Department of Pediatrics, Diabetology, Endocrinology and Nephrology, Medical University of Lodz, 91-738 Lodz, Poland

**Keywords:** c-peptide, diabetes, diabetes-associated complications, nephropathy, neuropathy

## Abstract

People with type 1 diabetes have an increased risk of developing microvascular complications, which have a negative impact on the quality of life and reduce life expectancy. Numerous studies in animals with experimental diabetes show that c-peptide supplementation exerts beneficial effects on diabetes-induced damage in peripheral nerves and kidneys. There is substantial evidence that c-peptide counteracts the detrimental changes caused by hyperglycemia at the cellular level, such as decreased activation of endothelial nitric oxide synthase and sodium potassium ATPase, and increase in formation of pro-inflammatory molecules mediated by nuclear factor kappa-light-chain-enhancer of activated B cells: cytokines, chemokines, cell adhesion molecules, vascular endothelial growth factor, and transforming growth factor beta. However, despite positive results from cell and animal studies, no successful c-peptide replacement therapies have been developed so far. Therefore, it is important to improve our understanding of the impact of c-peptide on the pathophysiology of microvascular complications to develop novel c-peptide-based treatments. This article aims to review current knowledge on the impact of c-peptide on diabetic neuro- and nephropathy and to evaluate its potential therapeutic role.

## 1. Introduction

Type 1 diabetes (T1D) is an autoimmune disease that causes destruction of insulin-producing beta cells of pancreas [1]. The discovery of insulin and its synthesis changed the natural course of the disease from a rapidly lethal to a chronic one. Currently, intensive insulin therapy allows patients to lead a long, quality life, providing they maintain a near-physiological glycemic control [2]. However, despite continuous progress in technology, patients with type 1 diabetes live on average about 10 years shorter than their peers without this disease [3]. This is mainly due to long-term complications of diabetes: micro- and macro-angiopathies, which eventually impair eyesight, central and peripheral nervous systems, kidney function, and cause increased cardiovascular risk.

Over the years, this accumulated vessel damage has been linked to chronic hyperglycemia and, lately, to increased glucose variability [4]. However, the exact pathophysiological links have not been completely resolved and there still might be unrecognized factors affecting the risk of T1D complications development. There is also a pressing need for new active agents or therapeutic strategies that might help to prevent or treat these disorders.

The connecting peptide of proinsulin, or c-peptide, has been investigated as a potential candidate to fill some gaps in the pathogenesis of T1D-related complications. Physiologically, it is a short 31-amino-acid sequence that joins A and B chains of proinsulin [5]. It facilitates the assembly and folding of the hormone in islet vehicles and is cleaved afterwards by endoproteases [6] to be secreted into the circulation in a 1:1 ratio to native insulin. Due to c-peptide’s high resistance to plasma peptidases it has a reasonably long and stable half-life (around 30 to 35 min) its concentration has been a long-standing marker of beta cell function and insulin secretion. 

There is, however, accumulating evidence that c-peptide is not just a by-product of insulin, but exhibits useful, if non-obvious biological properties [7]. These reports are especially important in view of current type 1 diabetes therapy regimens, which are centered on the delivery of exogenous insulin. This hormone is already folded and active, and therefore no c-peptide is co-administered via the subcutaneous route. Such formulation, however, creates a physiological gap between people with T1D and their healthy peers, which might contribute to the development of chronic complications.

Unfortunately, despite intensive research, there are many conflicting reports on the biological activity of c-peptide and its molecular targets. Similarly, the relationship of c-peptide with chronic T1D complications was also studied extensively; however, so far, no replacement therapies were shown to be successful in humans. Therefore, there is an ongoing need to decipher the function of c-peptide and its mechanism of action to determine whether or not, and how, it should be co-administered with insulin and facilitate translation efforts.

This work aimed to review current knowledge on c-peptide-activated pathways from the perspective of known mechanisms behind neuro- and nephropathy in T1D, with focus on the available evidence for c-peptide protective effect in animal and human studies In addition, we provide an overview of the most significant findings in the field and a summary of methodologies used in these studies as well as critically analyze the reasons behind the discrepancies between pre-clinical studies and human trials. 

## 2. Literature Search

We searched Pubmed using the following queries: “c-peptide and nephropathy and type 1 diabetes”, “c-peptide and retinopathy and type 1 diabetes”, “c-peptide and neuropathy and type 1 diabetes”, “c-peptide and microvascular complications and type 1 diabetes”, “c-peptide and macrovascular complications and type 1 diabetes”, “c-peptide replacement therapy and type 1 diabetes”, “c-peptide and Na^+^ K^+^ ATPase”, “c-peptide and eNOS”, “c-peptide and ERK”, and “c-peptide and GPR146”. We critically appraised the identified papers, excluding those published before 1990, in languages other than English and case reports. In the end, we chose 53 papers covering studies in cell lines (N = 10), animals (N = 20), and humans (N = 11), as well as relevant reviews (N = 12). The detailed flowchart of the literature search is given in Figure 1. 

## 3. Overview of Diabetic Microangiopathies

Diabetic neuropathy is a chronic complication that cumulatively affects up to 50% of patients with T1D [8]. Its most common presentation is chronic symmetrical length-dependent sensorimotor polyneuropathy, known as diabetic polyneuropathy (DPN), but the nerve damage can also manifest as autonomic neuropathy, diabetic radiculoplexopathy, various mononeuropathies and treatment-induced neuropathies. Diabetic neuropathy might affect all types of peripheral nerves—sensory, motor and autonomic—at various stages, which makes it extremely heterogenous from a clinical perspective. Currently, the disease is recognized by signs- and symptoms-based probability scales such as Toronto criteria [9]. 

Quantitatively, DPN is assessed by measurement of nerve conduction velocities, which are slowed in the affected patients. Clinically it is characterized by positive, i.e., new and unwanted symptoms that arise due to nerve damage, as well as negative ones mirroring nerve’s inability to perform its physiological functions. Positive symptoms include paresthesia (prickling, tingling, and ant-like sensations) and pain, while the negative ones are usually represented by numbness or a feeling of reduced sensation while walking. DPN is also one of the main risk factors of diabetic foot ulceration and consequently, amputation [10]. Currently, there are no effective treatments for DPN, as the condition is usually diagnosed late when nerve damage is already established [11]. Therefore, patients that develop DPN require complex, time- and resource-consuming multidisciplinary care. 

The other most common diabetes-related neuropathy is autonomic dysfunction, most often recognized as cardiovascular autonomic neuropathy (CAN). It manifests with tachycardia [12], QTi prolongation, lower heart-rate variability in ECG as well as orthostatic hypotension and reverse dipping in blood pressure measurements. Similarly to DPN, it needs to be managed as a multidimensional problem with many underlying factors and added risks. 

The most important risk factor for development of diabetic retinopathy appears to be (similar to other microangiopathies) poor long-term glycemic control evidenced by high HbA1c. Although DCCT and EDICT trials reported that some patients receiving intensified treatment also developed neuropathies [13], the perspective on optimal glycemic control has considerably evolved since then. However, as evidenced by large-scale cross-sectional studies, achieving and maintaining optimal glycemic control in T1D is challenging for many patients, especially youth [14]. This coincides with alarming reports of DPN being diagnosed in as much as 11% of children and adolescents [15] and emphasizes the need for developing additional preventive steps besides good metabolic control and healthy lifestyle.

Adverse impact of diabetes on nerves is generally attributed to prolonged hyperglycemia and resulting metabolic changes, dyslipidemia, microangiopathy, and chronic inflammation [8,16].

Early changes occurring in T1D models include decreased neural blood supply, increased activity of polyol pathways and impaired function of neural Na^+^/K^+^-ATPase and nitric oxide synthase (NOS), which is structurally reflected by axonal swelling [17]. They result in a potentially-reversible decrease in sensory and motor nerve conduction velocity. However, these changes are later aggravated by additional mechanisms: oxidative stress, chronic inflammation [16] and reduced expression of neurotrophic factors such as nerve growth factor (NGF) and insulin-like growth factor (IGF-1). These cause permanent structural changes such as axonal atrophy. However, the view on neuropathy development evolved considerably, especially in terms of differences according to type of diabetes. First, researchers observed some nodal and paranodal abnormalities that were unique to insulin- and c-peptide-deficient T1D and did not occur in hyperinsulinemic and c-peptide-abundant type 2 diabetes (T2D) [18]. These included lateralization of Na^+^ channels causing conduction blocks and resulting in disruption of ion-channel barriers and axoglial disjunction. Notably, these changes were demonstrated to be related mostly to impaired insulin signaling and not hyperglycemia itself [18]. Moreover, it was shown that the exact mechanisms of nerve damage differ between T1D and T2D, with deposition of advanced glycation end products in nerve extracellular matrix seems prevalent in T1D and microvascular or intraneural lipid depositions in T2D [19].Therefore, with current knowledge it is impossible to determine whether the differences in DPN pathology between T1D and T2D result from c-peptide deficiency or are mediated by other differentiating factors, e.g., lipid metabolism. However, many of the pathophysiological checkpoints ubiquitous in DPN pathophysiology can be affected by c-peptide, which shows promise as an active agent modifying the natural course of DPN.

Diabetic nephropathy, also known as diabetic kidney disease (DKD), is another chronic microvascular complication that is estimated to affect 25–40% of T1D patients during their lifetime. Advanced DKD is responsible for 30–50% of end-stage kidney disease in patients in the US [20] in need of renal replacement therapy [21] and also increases cardiovascular mortality.

DKD develops as a result of a series of pathophysiological events triggered by hyperglycemia that affects the function and structure of glomeruli. Initially, high blood glucose increases renal blood flow and glomerular filtration rate (GFR). This hyperfiltration is further aggravated by growth hormone, glucagon, and nitric oxide [22]. The resulting increase in intraglomerular and trans-capillary pressure as well as elevated mesangial matrix proliferation cause glomerulosclerosis. Moreover, disturbance of renal hemodynamics triggers the release of cytokines and growth factors that cause further damage and fibrosis of interstitial tissue. The effect is progressive decline in GFR, which eventually leads to chronic renal failure. 

Clinically, DKD remains asymptomatic for a long time [23] and its manifestation reflects renal failure. The symptoms reported by patents include foamy urine (indicating presence of protein), edema of feet and fatigue (suggesting hypoalbuminemia). DKD can then be divided into stages based on patient’s GFR, from prenephropathy (stage 1) to end-stage renal disease (stage 5) [24]. It is important that DKD may progress differently among patients, and in rapidly-progressing cases, some stages may be skipped altogether. 

So far, the main counter-measure to prevent DPN and DKD is achieving and maintaining strict glycemic control. However, even therapeutic success (>70% of time spent in glucose target range [25]) does not abolish the risk completely. This emphasizes the need to investigate therapeutic options to prevent the development of microvascular complications. One such treatment might include c-peptide administrations.

Its potential was hinted at by observational studies as early as in 1990. Winocour et al. [26] found out that among patients with long-standing T1D those with detectable post-prandial (>20 pmol/mL) c-peptide demonstrated lower prevalence of proliferative retinopathy than non-secretors (matched for age, diabetes duration, and BMI). However, results for DPN, CAN or DKD remained inconclusive. Moreover, after two years of observation, initial c-peptide status was no longer associated with any complications. Next, to define a potentially-relevant c-peptide concentration, a large cross-sectional study was performed by Kuhtreiber et al. [27]. In 324 patients, they assayed a fasting c-peptide and used multivariate logistic regression to test its association with collective presence of diabetes complications (neuropathy, nephropathy, retinopathy, or foot ulcers). In effect, they demonstrated that c-peptide ≥10 pmol/L was associated with protection from complications after adjustment for diabetes duration. However, these analyses did not adjust for the HbA1c concentrations. Hence it is not clear whether the observed effects were due to the specific action of c-peptide or lower HbA1c improved by high residual insulin production. Moreover, later studies demonstrated that such high concentrations are rarely achieved and maintained by T1D patients [28]. In terms of autonomic nerve dysfunction, a cross-sectional study by Ziegler et al. [29] showed that in T1D patients heart rate variability was significantly associated with glucagon-stimulated c-peptide secretion. Those with low c-peptide response present with blunted and lowered heart-rate variability, which is a characteristic of CAN. Importantly, these associations were preserved after adjustment for HbA1c and diabetes duration. Interestingly, in another arm of this study featuring patients with type 2 diabetes (and usually hyper-physiological concentrations of c-peptide), these associations were remarkably smaller.

Summing up, it is possible that preserved c-peptide secretion may offer protection from long-term diabetes complications. However, this is likely limited to the few patients with sufficiently-high c-peptide concentrations and to the time period when these concentrations are maintained. Eventually, c-peptide concentrations drop to clinically-insignificant levels and offer no visible protection against complications. On the other hand, high concentrations might be easily reached and preserved by c-peptide administration. Such treatment should be, however, based on investigation into c-peptide biological activities in vitro and in vivo.

## 4. Physiological Effects of C-Peptide and Its Potential Targets in Microvascular Complications

Multiple experimental studies on c-peptide replacement therapy in insulin and c-peptide deficient animals concluded that it exerts beneficial effects on nerve and kidney function.

In general, all projects first induced diabetes by different protocols and then supplemented the model animals with c-peptide for varying periods of time, to finally apply behavioral and laboratory tests and pathological examination to assess the progress of organ-specific complications.

Research on diabetic neuropathy showed that c-peptide supplementation diminishes nerve conduction slowing and functional deficits (thermal hyperalgesia [30], tactile allodynia), which are observed in non-supplemented animals with type 1-like diabetes [30,31,32,33]. These physiological improvements are mirrored in morphological and histological findings. In c-peptide treated animals, researchers observed significantly reduced frequency of axonal degeneration, myelinated fibers pathologies, Wallerian degeneration, paranodal swelling, axoglial disjunction, and paranodal demyelination [32] as well as improvements in myelinated fibers’ numbers, size, and axonal areas [34]. In terms of c-peptide influence on kidney function, it has been found to reduce glomerular hyperfiltration [35], microalbuminuria [35], and mesangial expansion [36]. Additionally, studies on streptozotocin-induced diabetes (STZ rats) showed that c-peptide infusion results in the improvement of glomerular function and in body weight gain and reduction of urinary sodium losses. 

Many candidate pathways were proposed to explain c-peptide biological activity (Figure 2) and they were studied using both cell-lines and animal models (Figure 3 and Figure 4).

The favorable effects of c-peptide on diabetic neuro- and nephropathy have been largely attributed to its ability to regulate Na^+^/K^+^ ATPase and endothelial nitric oxide synthase (eNOS) activity. Na^+^/K^+^ ATPase is a membrane ubiquitous enzyme, which moves sodium and potassium ions through cell membrane against its concentration gradients [37]. It maintains resting potential, regulates cellular volume, and transduces signals to regulate reactive oxygen species (ROS), intracellular Ca2^+^ and mitogen-activated protein kinase (MAPK) pathway. Reduced activity of this enzyme in peripheral nerves leads to inactivation of Na^+^ channels and intraaxonal Na^+^ accumulation at the node, which results in paranodal swelling [32]. C-peptide replacement for two months prevented paranodal swelling by 61% and reduced the neural Na^+^/K^+^ ATPase defect by 55% in Wor/BB rats [38] and, prevented a decrease in Na^+^/K^+^ ATPase activity in the sciatic nerve and granulation tissue in hyperglycemic conditions in STZ rats [39]. It is possible that c-peptide-stimulated increase in Na^+^/K^+^ ATPase activity prevents paranodal swelling and subsequent disruption of paranodal-axoglial junctions and in this way improves NCV in diabetic neuropathy. 

Similarly, numerous studies reported a stimulatory role of c-peptide on Na^+^/K^+^ ATPase activity in in vitro models of diabetic nephropathy [40,41]. The results of past research suggest that c-peptide treatment may alleviate glomerular hyperfiltration by inhibiting Na^+^ reabsorption in the proximal tubule via activation of Na^+^/K^+^ ATPase. It has been reported that c-peptide increases phosphorylation of the alpha subunit of Na^+^/K^+^ ATPase and in this way facilitates its function. C-peptide treatment induces an increase in the activity of Na^+^ pump and phosphorylation of the alpha subunit of Na^+^/K^+^ ATPase in rat medullary thick ascending limb of Henle’s loop [42]; however, this effect is not associated with an increase in the level of expression of the enzyme on cell membrane. In contrast, another study [43] demonstrated that c-peptide not only promotes phosphorylation of Na^+^/K^+^ ATPase alpha subunit, but also increases the basolateral membrane abundance of Na^+^/K^+^ ATPase alpha and beta subunits. Both of these studies showed that the effects of c-peptide treatment were abolished with a use of protein kinase C (PKC) inhibitor, which indicates an importance of this protein in the ability of c-peptide to exert its effects on Na^+^/K^+^ ATPase activity. The essential role of PKC was confirmed by a study [41] which showed that c-peptide treatment increases the activity of Na^+^/K^+^ ATPase as well as PKCε phosphorylation and extracellular signal-regulated kinase 1/2 (ERK1/2) activity in human renal tubular cells. In addition, it determined that treatment of this cell line with 1 nM of c-peptide increases DNA binding activity of zinc-finger E-box binding protein (ZEB) transcription factor. Since ZEB is involved in the expression of Na^+^/K^+^ ATPase alpha 1- subunit, it may explain, how c-peptide induces the increase in the activity of this enzyme. 

Impaired blood flow is another important factor influencing renal and nerve function in microvascular complications of diabetes. The progress of DN and DKD is characterized by a decrease in the activity of endothelial nitric oxide synthase (eNOS). eNOS is a member of a family of enzymes which catalyze the production of nitric oxide from L-arginine. It maintains endothelial homeostasis by regulating vascular tone and platelet activation. Reduced expression of eNOS was reported in glomerular [44] and cerebrovascular [45,46] endothelial cells of diabetic animals. Mice with eNOS deficit had increased levels of proinflammatory cytokines and more advanced pathophysiological changes in their kidney [47], whereas mice with a reduction of eNOS phosphorylation exhibited reduced cerebrovascular function and increased infract size [48]. Many studies showed that c-peptide is able to upregulate eNOS activity and act as a vasodilator. It has been demonstrated, in a model of diabetic neuropathy, that c-peptide treatment restores NOS activity and improves nerve microcirculation [49]. Furthermore, co-administration of NO synthase inhibitor markedly reduced the beneficial effects of c-peptide on NCV [49]. Conversely, at the early stages of diabetic nephropathy, c-peptide was shown to downregulate diabetes-induced elevated levels of eNOS and NO in the glomerulus and afferent arteriole. As a result, c-peptide prevented an initial increase in glomerular filtration rate (GFR), which occurs at the beginning of nephropathy. However, the mechanism by which c-peptide regulates eNOS activity in this system remains unclear. 

Pro-inflammatory environment, which is created due to chronic hyperglycemia contributes to the progress of diabetes-induced microvascular complications [50]. It was shown that inflammatory cells may accumulate in the glomeruli and interstitium of a kidney and in this way promote renal fibrosis [51]. In DPN, chronic inflammation was linked to peripheral nerve fiber damage and loss [16]. Multiple findings confirm that c-peptide is able to exert anti-inflammatory effects by regulating the activity of some components of pathways involved in inflammation including ERK1/2, c-Jun N terminal kinase (JNK), transforming growth factor β (TGF-β) and nuclear factor kappa-light-chain-enhancer of activated B cells (NF-κB). A study performed by Chima et.al [52] showed that c-peptide alleviates inflammation in rat kidney after hemorrhagic shock via reduction of ERK1/2 activity. The investigators also demonstrated that c-peptide treatment decreases the activity JNK and activating protein 1 (AP1). In contrast, another study [53] showed that c-peptide induces phosphorylation of ERK1/2 in a concentration dependent manner in renal tubular cells. The results of this study also indicated that c-peptide increases phosphorylation of JNK via ERK1/2 and induces Akt activity. They also showed that MEK1/2 inhibitor blocked the c-peptide effect on ERK1/2 phosphorylation. These conflicting data may suggest that c-peptide regulates ERK1/2 activity in multiple ways, potentially via different receptors. ERK1/2 mediated anti-inflammatory effects of C-peptide were confirmed by showing that c-peptide inhibits TGF-β via the activation of ERK in streptozotocin-induced diabetic mice [54]. TGF-β has been shown to trigger abnormal production of extracellular matrix (ECM) in the glomeruli of diabetic rats thus contributing to glomerular sclerosis and interstitial tubular damage. The effects of c-peptide could potentially ameliorate these events.

The ability of c-peptide to modulate inflammatory pathways in a kidney was also recently reported by Alves et.al. They showed that STZ-diabetic mice treated with c-peptide exhibited reduction in the urinary levels of anti-inflammatory IL4 and IL10 and proinflammatory IL17 and TGF-α as well as an increase in IL10 gene expression and a decrease in TGF-α gene expression when compared to diabetic non-treated group. This suggests that c-peptide may counter-regulate IL10 and TGF-α and modulate pro-and anti-inflammatory pathways and in this way attenuate kidney inflammation [55].

C-peptide may exert its anti-inflammatory effects by influencing the activity of NF-κB, which is a pivotal mediator of inflammatory responses. There is evidence that c-peptide reduces NF-κB activation, affects vascular cell adhesion molecule 1 (VCAM-1) expression and monocyte chemoattractant protein-1 (MCP-1) and interleukin 8 (IL-8) secretion in human aortic endothelial cells (HAEC) [56].

Another important property of c-peptide is its ability to promote cell survival and proliferation. The results of many studies show that hyperglycemia promotes apoptosis of various cell types including renal and neural cells. A study performed by Al-Rasheed et al. [57] showed that c-peptide stimulates proliferation of opossum proximal tubular kidney cells. It does so by increasing the activity of ERK and phosphatidylinositol 3-kinase (PI3K). C-peptide-induced proliferation was inhibited by treatment of cells with wortmannin and pertussis toxin. It suggests that proliferation of these cells is triggered via activation of PI3K and the signal is transduced by G-protein coupled receptor. Proliferative effects of c-peptide has been demonstrated on human renal mesangial cells [58]. The study indicated involvement of SRC-kinase, PI3K, and ERK1/2. C-peptide-induced proliferation may potentially counteract progressive kidney damage caused by hyperglycemia by stimulating these pathways. However, it is still uncertain whether the pro-proliferative effects of c-peptide on kidney cells are harmful or beneficial. 

Moreover, another study [59] showed that c-peptide prevents spatial learning and memory deficits as well as hippocampal neuronal loss in BB/Wor rats. It is a result of inhibiting the increased expression of Bax and active caspase 3, hence preventing apoptosis. Along the lines, c-peptide was found to affect other possibly important pathophysiological components involved in DN development: IGF-1 receptor expression, NGFR TrkA and TrkC expression in dorsal root ganglia and loss of substance P and CGRP [30,60]. Importantly, these effects occurred despite no significant changes in blood glucose levels [31]. The expression of IGF-1 and NGF was shown to be reduced in diabetic rats and connected to progressive axonal degeneration. C-peptide may promote neurotrophic support and promote the synthesis of neurofilaments in this way affecting axonal size. 

It remains unclear which receptors transduce c-peptide’s signal. In 2013, Yosten et al. [61]. determined that a knockdown of an orphan G-protein-coupled receptor (GPCR)—GPR146, blocks c-peptide-induced cFos expression in KATOIII cells. Since then, GPR146 remains the main candidate for c-peptide receptor. However, recently, another group claimed there is no indication that c-peptide acts through any of the GPCRs expressed in tested cell lines, including GPR146 [62]. At the time of this review, this controversy remains unsolved.

## 5. Interventions in Humans

We managed to identify the total of five clinical trials that explored the effects of c-peptide supplementation in humans (Figure 5). The first c-peptide interventions were conducted in 1992 [63] and 1993 [64] by Johnsson et al. Those were small randomized, double-blind pilot studies investigating the effect of c-peptide on renal function and (in case of the second study) blood–retina barrier. In the first study c-peptide was administered intravenously for 1 h while in the second one the biosynthetic c-peptide was mixed with equimolar amount of insulin and administered subcutaneously for four weeks. Both studies included a control group of T1D patients receiving 0.9% NaCl infusion (in the first one) and just insulin in the second study. As a result, both studies reported a significant decrease in glomerular hyperfiltration reflected by GFR and albuminuria without significant impacts on effect on glycemic control parameters such as blood glucose, HbA1c, and fructosamine. The effect on blood–retina barrier permeability was borderline-significant; however, it should be treated with caution as c-peptide group presented with significantly higher blood–retina leakage than control. Importantly, no adverse effects were observed during c-peptide administration.

Initial positive results for short-term c-peptide substitution motivated Johansson et al. to investigate if they can be maintained during long-term administration. A double-blinded cross-over randomized controlled trial was performed in which normotensive T1D adult patients with microalbuminuria were given subcutaneous injections of c-peptide or placebo for three months. In effect, a considerable decrease in both albumin excretion (by 40%) and albumin/creatinine ration (~30%) was observed during c-peptide treatment. However, no significant change in GFR was observed. On the other hand, this study group presented lower GFR at baseline (mean 108 ± 3 mL/min/1.73 m^2^) than participants from previous studies. It is therefore likely that those patients already passed the hyperfiltration stage window for c-peptide action. Following on the previous study, the blood–retina barrier leakage was also assessed but revealed no significant effect of c-peptide administration.

The pooled effect of c-peptide supplementation on GFR was summarized in a systematic review and meta-analysis by Shaw et al. [65]. They concluded that, collectively, c-peptide administration had no statistically-significant impact on GFR. However, these conclusions seem to not be backed up by statistical analysis. In their work, Shaw et al. compared only end-point measures of GFR between intervention and control groups and disregarded baseline values. Thus, the difference in GFR change experienced by treated and not treated patients may have been lost.

Overall, c-peptide administration demonstrated a positive but clinically-limited effects on the nephropathy-associated functional abnormalities. The decrease in glomerular hyperfiltration and albumin excretion hint for potential clinical utility. 

Interestingly, the already-mentioned study by Johnsson et al. also provided cautious but promising results for DPN. C-peptide administration was associated with statistically-significant increase in heart rate variability (initially reduced in N = 12) and lowering of thermal and vibration sensory threshold (abnormal at baseline in N = 6). 

These tentative results coupled with promising findings in animal studies resulted in a few double-blinded randomized clinical trials dedicated to DPN being performed in humans. The first, exploratory one [66], was carried out in 46 patients with T1D without DPN symptoms and 15 healthy volunteers. The protocol included three months of treatment with either c-peptide or placebo in four subcutaneous injections. Such replacement therapy restored serum c-peptide concentration to a physiological level, but the testing was done relatively soon (~3 h) after injection. There was no significant HbA1c change in either group during the treatment, however, between-group comparison in terms of HbA1c difference was not performed. Importantly, no adverse reactions or events related to c-peptide administration were observed. Clinically, neither patients with T1D nor healthy controls reported any DPN symptoms, but those with T1D patients presented reduced sensory and motor nerve conduction velocities (SNCV, MNCV) in the sural and peroneal nerves, which was consistent with subclinical DPN. C-peptide administration improved SNCV significantly throughout 12 weeks of treatment but only by ~5% and slightly improved vibration threshold in comparison to no improvement in healthy subjects. MNCV changes were transient and similar to those observed in control group. Improvements of both SNCV and the vibration threshold in patients with subclinical DPN suggest that it might act in very early stages. 

Another study [67] picked up on these promising findings and recruited patients with T1D and signs and symptoms of DPN, with SNV < −1.5 SD from body-weight-corrected reference. The patients were randomized to receive placebo control or c-peptide in either low (physiological) or high dose. The supplementation was carried out for six months, with neurological examination and neurophysiological measurements performed before and after. In analysis, low and high dose groups were pooled because there were no significant differences between them in response.

As a primary endpoint, the study demonstrated significant improvement in SNCV in c-peptide treated group, but placebo-treated group also experienced similar benefit. Nevertheless, the proportion of responders with improvement in SNCV below 1 m/s was significantly greater in the intervention groups. From a clinical perspective, vibration perception threshold and neurological examination score were also significantly improved by c-peptide supplementation. A secondary analysis also hinted that among those with better SNCV at baseline, c-peptide administration produced significantly greater improvement than placebo. MNCV deteriorated during six months in all study groups, without any apparent effect of c-peptide. However, this report was accompanied by a breach of publication ethics that should be recalled. Two of the paper’s first authors were (at the time of publication) patent holders for therapeutic use of c-peptide [68], which constitutes an important conflict of interest that was not disclosed. On the other hand, the publication described the proceedings and analysis well enough that reported data seem reliable.

Following these results, the (partly the same) group attempted to improve on the formulation of c-peptide. Both of the above-mentioned trials used native c-peptide administered subcutaneously four times a day as an active agent. Following the results from animal studies, the next trial tested the PEG-ylated form administered weekly in high (2.4 mg) and low (0.8 mg) dose [69]. This time, the authors disclosed having financial relationships with Cebix Incorporated, a company holding patent for PEG-ylated c-peptide. However, the trial publication failed to name Cebix as the trial sponsor. This trial [69] recruited T1D patients with mild-to-moderate peripheral neuropathy, blinded and randomized them while stratifying for HbA1c, diabetes duration and center location. Moreover, the planned follow-up was 13 months, which is the longest reported time of treatment. The outcomes were assessed after 26 and 52 weeks of supplementation and included comprehensive array of DPN signs and symptoms: sensory and motor nerve conduction, vibration perception thresholds, modified Toronto Clinical Neuropathy Score (TCNS) assessment, reports for pain, sexual function, and erectile dysfunctions. 

The results, however, were comparable with the previous studies. SNCV improvement was observed in all study groups with no significant difference between placebo- and c-peptide- treated patients. Vibration perception threshold decreased in both groups supplemented with c-peptide with insignificant difference between the doses. Other neuropathy features (including MNCV and TCNS) did not improve with c-peptide administration. 

Overall, human trials of c-peptide supplementation produced rather underwhelming results despite testing various doses and different follow-up times. The only endpoint showing consistent improvement was vibration perception threshold. However, the reliability of this method is controversial due to poor standardization of measurement and its vulnerability to confounding factors, such as age and height [70]. Still, it is an important symptom that warrants further studies into possible utility of c-peptide replacement therapy. It could lead to developing a therapy that aims to prevent or even reverse previously mentioned DN symptoms. The time for intervention, as demonstrated by Ekberg et al. [66], might also be an important point to consider in inventing effective pharmaceutical therapy of DN. The major limitation of these findings emerge from the fact that they all come from one center—Karolinska Institute, which may create problems with the reproducibility of the results. In addition, the previously-discussed transparency issues seem to further undermine the clinical validity of the findings. Should further efforts be made to bring c-peptide into the market and clinical practice, they should be thoroughly scrutinized for transparency. 

## 6. Translational Gap

Overall, despite very promising results in pre-clinical animal trials, in humans the observed effects of c-peptide administration on chronic diabetes complications are much weaker. The exact reasons for these discrepancies were not identified but can be speculated to arise from either technical or methodological weaknesses of previous studies.

First of all, there is wide diversity in terms of chemical purity of c-peptide used in pre-clinical research. While most protocols reported to have used up to 98%-pure c-peptide, these qualities were given in terms of mass percentage, which does not reflect molar purity. As revealed by Pinger et al. [71], even 2 mass % of impurities may translate in case of Fe add-ons into 50:50 molar ratio in the administered substance. This is unlikely to have affected (at least that heavily) human-based trials where purity of administered compounds is much more stringent—however, it potentially explains differences between animal studies and hints that additional substances might modulate c-peptide activity.

This was indeed demonstrated by Liu et al. [72], who reported that c-peptide biological effects are heavily reliant on the presence of albumin and zinc ions. While both should be abundantly present in circulation of patients receiving c-peptide, it cannot be excluded that other co-factors present or absent in human circulation might affect c-peptide action.

Furthermore, one must consider that animal models might not accurately represent diabetes-related complications encountered in humans with T1D. In most reviewed cases, diabetes was induced pharmacologically by streptozocin administration, which caused abrupt and preferential destruction of beta cells. This method, albeit effective, does not reproduce complex autoimmune environment that over the years causes the development of T1D and possibly continues to affect the patient in later life. Secondly, in c-peptide-related studies the animals did not receive physiological supplementation of insulin, which is the case in humans. In most cases, the insulin was titrated to achieve and maintain hyperglycemia ~20 mmol/L. Biologically, such environment is drastically different from that of humans with established diabetes, even those who do not achieve good metabolic control. As such, the damage of nerves and kidneys observed in rodents develops differently than in humans—faster, with more dependency on hyperglycemia. In such insulin-deficient environment, the insulin-like properties of c-peptide might be more pronounced and strengthen observed positive effects. To determine the unique effect of c-peptide not affected by hyperglycemia, animals studies would have to be carried out on an animal model of microvascular complications with maintained euglycemia, which, according to our knowledge, has not yet been developed. On the other hand, c-peptide does not seem to exert much effect on blood glucose levels; it does not counteract hyperglycemia in T2D despite its abundant presence in the serum. It is possible that insulin and c-peptide resistance occur simultaneously in this type of diabetes; however, taking into consideration that insulin treatment is sufficient to neutralize blood glucose levels, this scenario appears unlikely. Moreover, the animal and human studies differ also in the time of intervention. The former targeted mostly the acute phase of organ damage while clinical trials recruited mostly patients with established (even if early-stage) neuro or nephropathy. It might be informative to observe the effects of joined c-peptide and insulin administration in 1:1 molar ratio started at diabetes onset. Such design might be feasible with modern insulin pumps, as dual-hormone pumps (administering insulin and glucagon) are already in development.

Additional investigation should also focus on the diabetes complications themselves—to elucidate more precise mechanisms of their development in humans, identify the most intervention-sensitive periods and possibly design new ways to measure the outcomes. The latter seems especially important in neuropathy, where even SNCV—which is measured objectively and considered one of reference methods to assess neuropathy severity—was demonstrated to improve in placebo group during the latest trial [60].

Finally, priority should be given to identification of receptor mediating c-peptide action. It might uncover significant differences between receptor structure between humans and rodents traditional factors affecting c-peptide signaling in human cells specifically. Furthermore, knowledge about receptor structure might help design more effective agonists to boost native c-peptide effects. With recent works debating the role of GPR146 receptor, there is hope that these details might be finally uncovered and utilized.

## 7. Conclusions

C-peptide has come a long way from being considered an inactive cellular by-product to acknowledgment as a biologically-active molecule. Many of its observed properties, as well as pre-clinical research suggests c-peptide might be utilized in prevention or treatment of chronic complications in T1D, with neuro- and nephropathy being the most attractive targets. However, so far the clinical trials have demonstrated only no successes at bench-to-bedside implementation. Observed effects, while present, were often small and pertained only to a few chosen parameters as opposed to expected global, clinically-meaningful changes. The fact that all clinical trials on DPN which were to some extent successful were carried out at one center makes it more difficult to assess the reliability of the obtained results. The lack of success in application of c-peptide treatment to patients might result from our incomplete understanding of c-peptide receptor interactions, imperfect representation of complications by animal models, or finally ineffective timing for the interventions. As a promising molecule in diabetology, c-peptide certainly deserves more in-depth research before it can be effectively translated into a second wave of clinical trials.

## Figures and Tables

**Figure 1 ijms-21-09723-f001:**
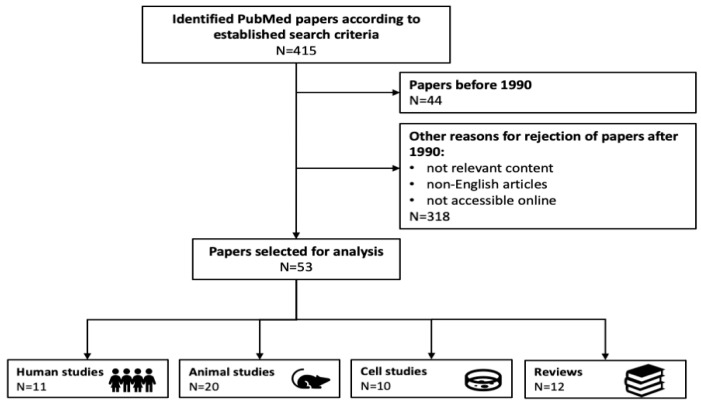
Flowchart illustrating selection criteria for analyzed papers. In total, there were 53 papers selected for the analysis.

**Figure 2 ijms-21-09723-f002:**
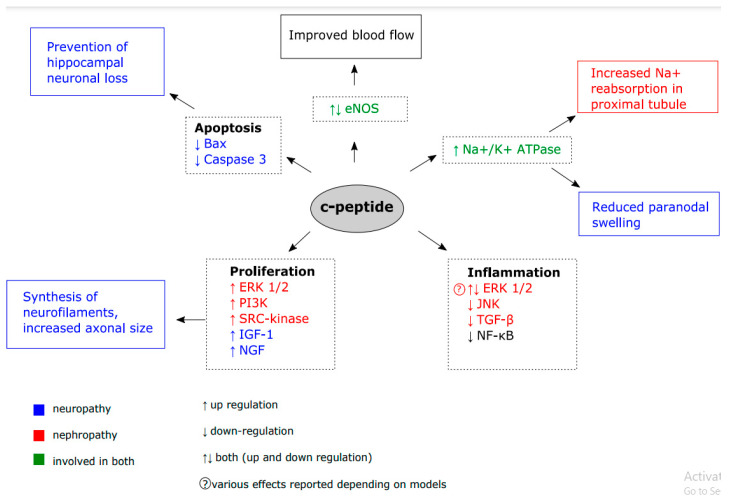
C-peptide treatment affects the function of cellular components involved in diabetic neuropathy (blue), diabetic nephropathy (orange) or both of these complications (green). As a result, C-peptide exerts beneficial effects on cells by reducing inflammation and apoptosis, improving blood flow, promoting proliferation and Na^+^ reabsorption in proximal tubule.

**Figure 3 ijms-21-09723-f003:**
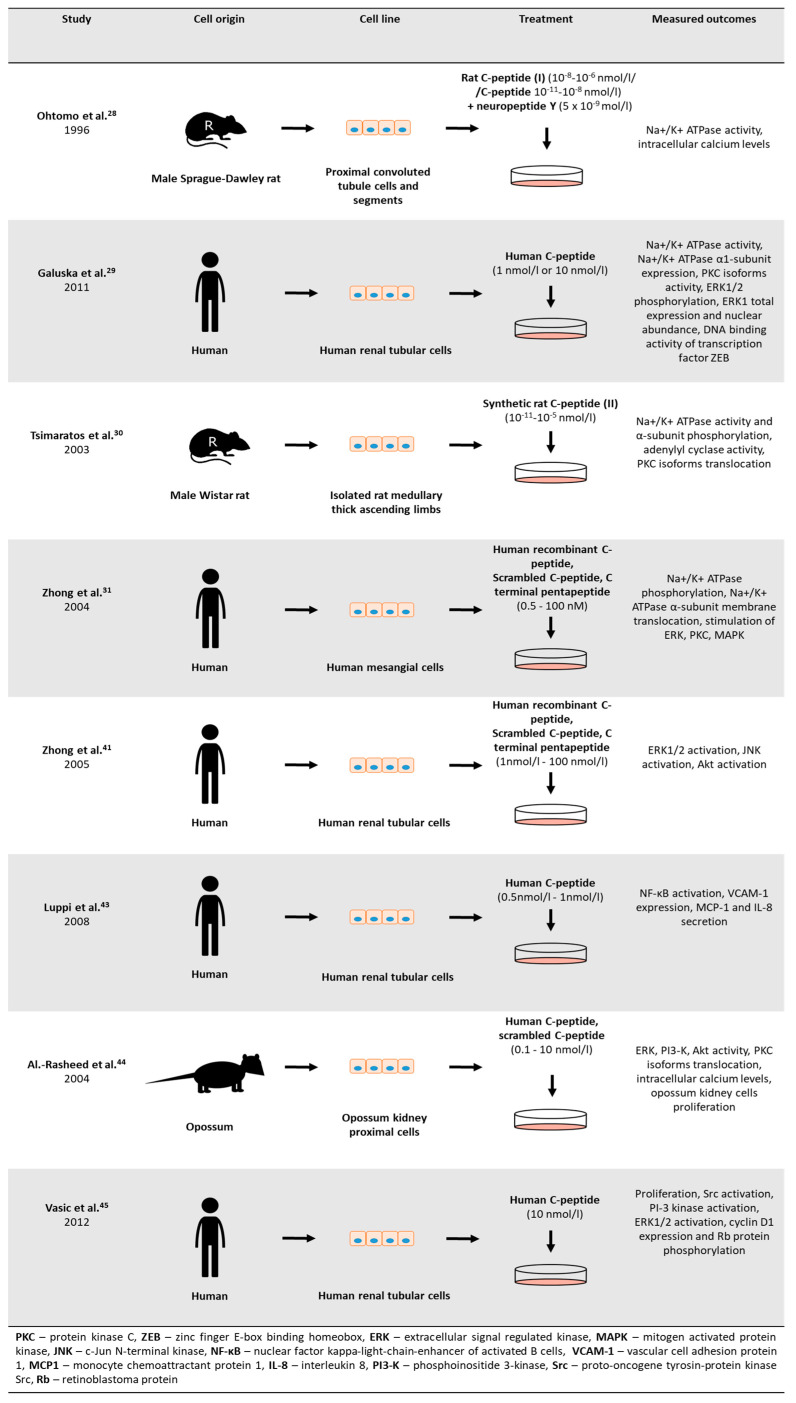
Study design of in vitro studies reviewed in this paper.

**Figure 4 ijms-21-09723-f004:**
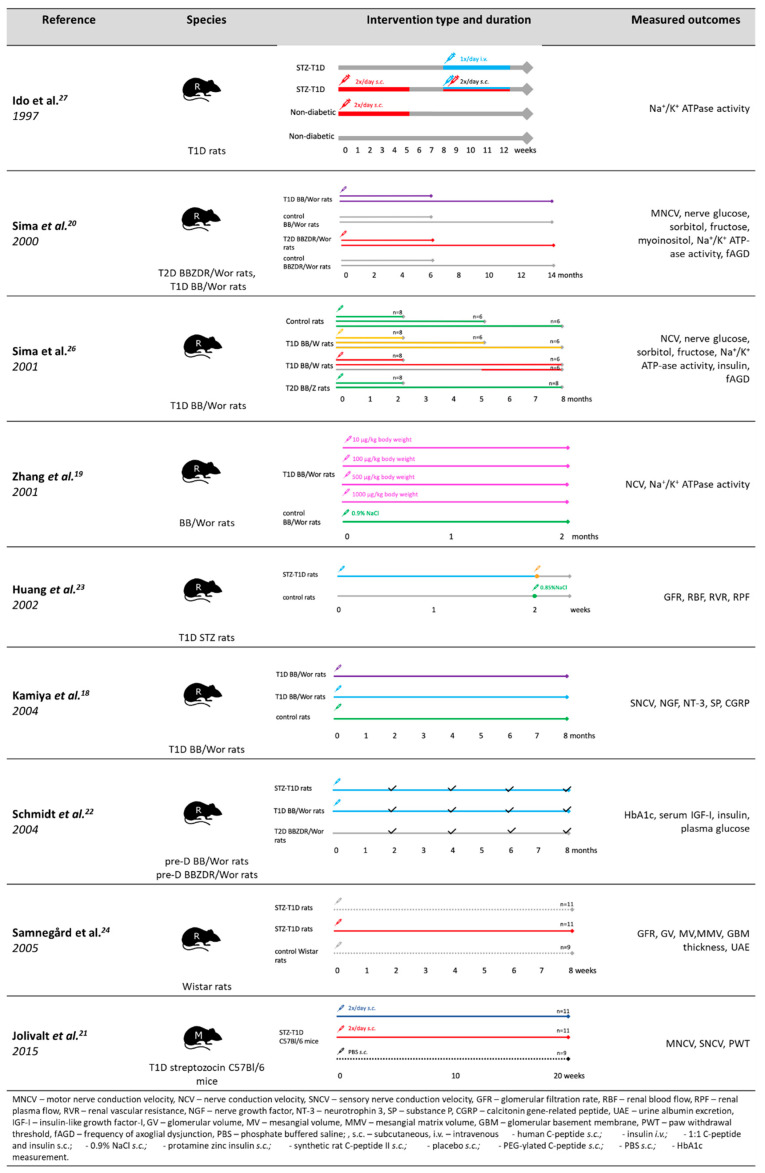
Study design of animal studies reviewed in this paper.

**Figure 5 ijms-21-09723-f005:**
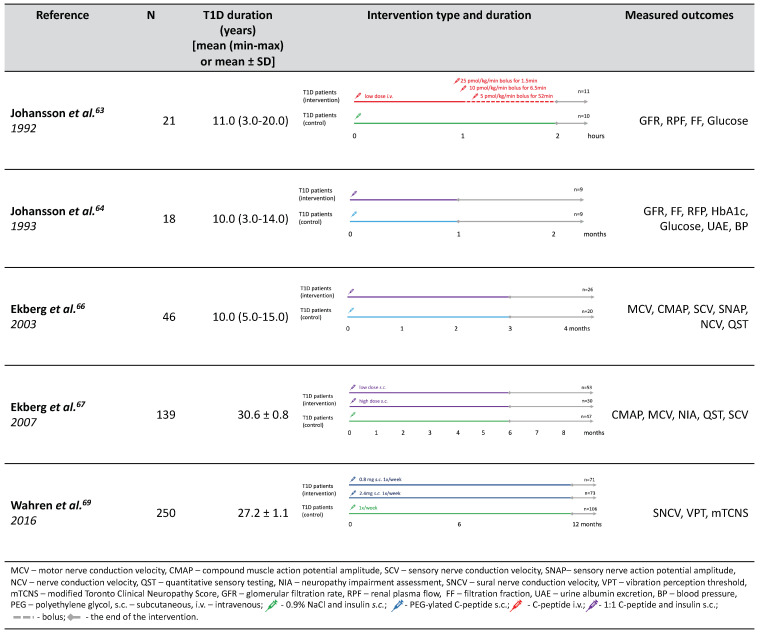
Study design of interventions in humans.

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
