# Peer review of "Biological Activity of c-Peptide in Microvascular Complications of Type 1 Diabetes—Time for Translational Studies or Back to the Basics?"

_ijms, 2020, doi:10.3390/ijms21249723_

Round 1

Reviewer 1 Report

Ryk comprehensively review the literature regarding cell, animal model and human studies of the effects of C-peptide on microvascular complications of type 1 diabetes, specifically neuropathy and nephropathy.

The review is clearly written, with a clear message and balanced discussion of the literature. The figures and tables are very helpful for understanding the design and major findings of animal model studies and clinical trials. The authors provide an important perspective on the gap in translation of findings from animal models to human trials and address future research in the field.

While a wide body of the literature is covered, the authors could add a recent publication to the review when discussing the effects of C-peptide on kidney inflammation (p. 6,7), discussing the Alves et al study in Mol. Biol Rep 47:721 (2020).

Author Response

Point 1:While a wide body of the literature is covered, the authors could add a recent publication to the review when discussing the effects of C-peptide on kidney inflammation (p. 6,7), discussing the Alves et al study in Mol. Biol Rep 47:721 (2020).

Response 1. We would like to thank the reviewer for this favorable opinion. We are also grateful for pointing to us one of the most recent works in the field. According to your suggestion, we added insights from Alves et al. in the appropriate section to bring it up to date.

Reviewer 2 Report

The authors feel enthusiastic about C-peptide based on animal and cell studies but the human data is particularly unconvincing. Although the authors were critical of the negative conclusions of the PLOS one review on nephropathy  they have not been particularly critical of the human data on neuropathy. I am not surprised it is now more than 28 years since the first human paper was published and there has been no therapeutic product.

The problems with the human data are 1. It almost all comes from one centre-The Karolinska Institute and 2 of the authors have a patent on C peptide which was not declared in any of the papers. To my mind this should  lead to retraction of at least the Diabetes Care paperof 2007 and probably the Diabetes paper.  The 2007 paper showed arbitrary post hoc  data selection to get a positive result when there was none overall. At least the 2017 paper mentions it was funded by CEbix who makes the C peptide (although this was not stated) and in this paper the results for sural nerve sensory velocities showed no difference at all between active and placebo. The findings on vibration perception are  much less objective than conduction velocities and should be treated with suspicion.

The table on human studies has misnumbered references and wrong data in the table which needs correction.

 I would disagree with the authors that  C-peptide is a promising molecule from the human data and I wouldn't suggest at present more trials need to be done.

Author Response

Point 1: Although the authors were critical of the negative conclusions of the PLOS one review on nephropathy they have not been particularly critical of the human data on neuropathy.I am not surprised it is now more than 28 years since the first human paper was published and there has been no therapeutic product. The problems with the human data are 1. It almost all comes from one centre-The Karolinska Institute and 2 of the authors have a patent on C peptide which was not declared in any of the papers. To my mind this should  lead to retraction of at least the Diabetes Care paperof 2007 and probably the Diabetes paper.  The 2007 paper showed arbitrary post hoc  data selection to get a positive result when there was none overall. At least the 2017 paper mentions it was funded by CEbix who makes the C peptide (although this was not stated) and in this paper the results for sural nerve sensory velocities showed no difference at all between active and placebo. The findings on vibration perception are  much less objective than conduction velocities and should be treated with suspicion.

Response 1. We would like to thank the reviewer for sound and meaningful feedback. Indeed, in our review we focused on the critical assessment of findings in nephropathy published in the PLOS ONE analysis. This was due to the fact that our field of expertise covers, among others, biostatistics and the findings of that papers raised our concern. On the other hand, we clearly missed the important issue of conflict of interest behind the reported findings in neuropathy area, as well as potential problems with reproducibility of single centre research. Therefore, we expanded the relevant part of our review to add sound critique of the clinical trials investigating role of c-peptide in diabetic neuropathy.

Given the additional details concerning c-peptide effects in a clinical setting, we modified our conclusions to acknowledge the shortcomings and disappointing results of past clinical trials. However, we still believe that c-peptide could be considered a promising molecule if more detailed and focused basic research is performed to determine the mechanism of its biological action. Due to the current lack of established secondary structure of c-peptide, unknown route of action and cellular pathways dependent on its action it would be unwise to disregard a bona fide hormone from further investigations. We therefore suggest in our work that  progress on c-peptide may be made solely by taking a step back, obtaining more quality evidence from in vitro and animal studies, and then pushing for another round of well-planned translational efforts.

Point 2: The table on human studies has misnumbered references and wrong data in the table which needs correction.

Response 2. The misnumbered references in the mentioned table were corrected.

Reviewer 3 Report

Biological activity of C peptide.

This is an interesting review. There was so much excitement when Proinsulin became available in Clinical practice but although there were suggestions that the addition of C peptide might have lipid renal and neurological benefit the company withdrew proinsulin following unexplained deaths.  The introduction is concise 

I was unable to find Ref 12 in Pub Med and wonder where there is good evidence that patients with tight control still develop neuropathy, The DCCT trial certainly showed that some tight control subjects developed neuropathy but it is uncertain how good their control.  Certainly HbA1c was the most important risk factor even after 20 years (DCCT/EDICT). Transplantation of pancreas and Kidney results in improvement in neuropathy and regeneration of small nerve fibres (Azmi S et al 2020 Expert opinion Pharmacotherapy) but of course C Peptide is also replaced.

Structural changes unique to Type 1 diabetes, Lines 97-104 needs a reference. Certainly, disturbance in lipid metabolism is usually more profound in Type 2 diabetes and affects nerve structure but is C peptide involved? The article by Jende JME et al  Annals in Neurology 588-598 2018 of interest.

DKD line 109 and following , Ref 17 is 2003 a more up to date reference worth while. The understanding of diabetic renal disease has progressed.

The article by Winacour et al ref21 suggests that residual C Peptide is associated with better glycaemic control and many newer studies have shown this.

Line 135 Reasonable to state that the study showed a very significant association between C Peptide and glycaemic control, Perhaps C peptide is only a marker for endogenous insulin?

Kamiya et al in 2006 Diabetes 55 3581 showed that C peptide reverses nociceptive neuropathy in diabetic rats but as the authors say C Peptide has many insulin like effects and the control group of rats were untreated hyperglycaemic animals  The article does not make it clear that in ref 28 it is untreated diabetic mice so again is it the effect of C peptide or the insulin like effect of C Peptide.

The paragraph starting line 174, again the Authors do not make it clear that insulin may have similar effects. Ref 36 Insulin was not used as control

Ref 44  No insulin comparator

The clinical section is concise and as the Authors comment the evidence of benefit from the addition of C Peptide is underwhelming.

In conclusion the preclinical studies section would benefit from a distinction being made between effect of C peptide and C peptide in comparison to insulin as perhaps most of the C Peptide effect is through its insulin action. This should be explored further since Patients without C Peptide have more difficulty in controlling their diabetes.  There is so much evidence that complications of diabetes can be prevented by good glycaemic control Patients with no C Peptide struggle to achieve good control and perhaps these patients would benefit from the addition of C Peptide?

A very interesting and well written review,

Author Response

Point 1: I was unable to find Ref 12 in Pub Med and wonder where there is good evidence that patients with tight control still develop neuropathy, The DCCT trial certainly showed that some tight control subjects developed neuropathy but it is uncertain how good their control.  Certainly HbA1c was the most important risk factor even after 20 years (DCCT/EDICT). Transplantation of pancreas and Kidney results in improvement in neuropathy and regeneration of small nerve fibres (Azmi S et al 2020 Expert opinion Pharmacotherapy) but of course C Peptide is also replaced. The effect may be more subtle or qualitatively different from those of typical complications of hyperglycemia. Given that it is a very challenging model to identify sublinical, unexpected morphological or functional alterations of patients devoid of c-peptide but good metabolic control for a long period of time, this area remains mostly unexplored and in need of a comprehensive, well-planned study.

Response1: We would like to thank the reviewer for meaningful feedback. We acknowledge that diabetes therapy has progressed so much over the years that current definition of tight diabetes control might differ from the one applied to DCCT/EDIC cohort. We agree that poor long-term glycemic control is still the most important risk factors for developing neuropathy and maintaining optimal HbA1c levels would be the best solution to decrease the risk. However, epidemiological data show that DPN is diagnosed in 11% of children and adolescents (Moser et.al 2013 Diabetes Res Clin Pract) which indicates that maintaining optimal glycemic control in T1D is challenging for many patients and there is a need for developing additional preventive solutions.

We also consciously decided not to review the effect of joined pancreas and kidney transplantation on complications exactly for the reason given by the Reviewer and added this information to the manuscript. Although the procedure demonstrated benefits, they cannot be attributed solely either to HbA1c improvement or c-peptide replacement.

Point 2: Structural changes unique to Type 1 diabetes, Lines 97-104 needs a reference. Certainly, disturbance in lipid metabolism is usually more profound in Type 2 diabetes and affects nerve structure but is C peptide involved? The article by Jende JME et al  Annals in Neurology 588-598 2018 of interest.

Response 2: We added the omitted reference to the study that demonstrated pathophysiological changes in neuropathy unique to type 1 diabetes. The article suggested by the reviewer is also an important report presenting findings from magnetic resonance neurography. We include those findings in the revised version of manuscript as supplementing pathological observations.

Point 3: DKD line 109 and following , Ref 17 is 2003 a more up to date reference worth while. The understanding of diabetic renal disease has progressed.

Response 3: According to your suggestion we added a more recent paper as a reference for this paragraph: Vikram and Vasanath Rao et al. 2019 Diabetic nephropathy: An update on pathogenesiss and drug development.

Point 4: Line 135 Reasonable to state that the study showed a very significant association between C Peptide and glycaemic control, Perhaps C peptide is only a marker for endogenous insulin?

Response 4: We agree that in many observations residual c-peptide secretion was associated with better glycemic control. This makes it difficult to decide whether the observed decrease in the prevalence of diabetes-related complication is attributable to lower HbA1c (achieved with residual insulin secretion) or c-peptide biological action. This can only be estimated by statistical adjustment of each effect in multivariate models. In the study by Kuhtreiber et al, this was not done (their model adjusted only for diabetes duration), which is an important limitation. We incorporated this point into the discussion, however it is unlikely to change the logic of the paper as “potentially-protective” levels of c-peptide secretion were later demonstrated as achieved in extremely rare cases.

Point 5: Kamiya et al in 2006 Diabetes 55 3581 showed that C peptide reverses nociceptive neuropathy in diabetic rats but as the authors say C Peptide has many insulin like effects and the control group of rats were untreated hyperglycaemic animals  The article does not make it clear that in ref 28 it is untreated diabetic mice so again is it the effect of C peptide or the insulin like effect of C Peptide.

The paragraph starting line 174, again the Authors do not make it clear that insulin may have similar effects. Ref 36 Insulin was not used as control

Ref 44  No insulin comparator

In conclusion the preclinical studies section would benefit from a distinction being made between effect of C peptide and C peptide in comparison to insulin as perhaps most of the C Peptide effect is through its insulin action.

Response 5: Thank you for this feedback. In most  animal models of diabetes/hyperglycemia, insulin was administered to both c-peptide-treated and untreated groups. However, the insulin supplementation was kept insufficient and targeted at maintaining persistent hyperglycemia. It is therefore possible that at least some of effects observed in c-peptide-treated animals were due to activation of insulin-dependent pathways (different than glucose utilization) by c-peptide. To determine the unique effect of c-peptide, it would be necessary to include also a third group treated with enough insulin to achieve and maintain euglycemia. Unfortunately, to our knowledge this was not done in any of the reviewed studies. Moreover, this might not even be feasible as persistent hyperglycemia is one of the main mechanisms leading to development of microangiopathies in animal models of diabetes. The solution might be creating new animal model of vascular complications that are developed in the long term and only in a proportion of animals to better reflect the human conditions. We address this point in the later section of the manuscript while investigating the possible reasons for the translational gap between animal and human studies. However, to emphasize that we agree with the Reviewer`s reservation, we expanded the relevant section.

Point 6: This should be explored further since Patients without C Peptide have more difficulty in controlling their diabetes. There is so much evidence that complications of diabetes can be prevented by good glycaemic control Patients with no C Peptide struggle to achieve good control and perhaps these patients would benefit from the addition of C Peptide?

Response 6: This is an interesting and worthy point to consider that c-peptide secretion might affect the patient`s glycemic control and indirectly impact the risk of developing complications. If the c-peptide has clinically-relevant biological activity (despite being marker of endogenous insulin secretion), its addition to standard insulin therapy might demonstrate a benefit on glycemic variability even if long-term complication-centered observation was impossible. As most human trials of c-peptide were performed in the past when continuous glucose monitoring was not prevalent, this is currently unknown but worth exploring.

Reviewer 4 Report

The authors provide an outstanding review paper about the actions of c-peptide in microvascular complications of T1DM. My concerns:

  1. The Title is misleading as it is. It must be ameliorated.
  2. The authors are kindly requested to formulate one or more concrete aims, and they should state them in the last paragraph of the introduction.
  3. The search strategy should be described in detail in Methods

Author Response

Point 1: The Title is misleading as it is. It must be ameliorated.

Response 1: We would like to thank the reviewer for your kind reception of our study and your suggestions. They were very helpful and helped us improve the manuscript`s content and clarity. Detailed answers to each remark are provided below:

The title of the manuscript has been changed as suggested.

Point 2: The authors are kindly requested to formulate one or more concrete aims, and they should state them in the last paragraph of the introduction.

Response 2: The aims have been formulated and added at the end of the Introduction section:

  • to review current knowledge on c-peptide-activated pathways from the perspective of known mechanisms behind neuro- and nephropathy in T1D, with focus on the available evidence for c-peptide protective effect in animal and human studies
  • to provide an overview of the most significant findings in the field and a summary of methodologies used in these studies
  • to critically analyze the reasons behind the discrepancies between pre-clinical studies and human trials

Point 3:The search strategy should be described in detail in Methods

Response 3: The search strategy paragraph has been added to the Methods section as well as a flowchart illustrating the process, as suggested. We did not follow the PRISMA guidelines as this manuscript is not a systematic review.

Round 2

Reviewer 2 Report

The reference to Johahnnson 1992 ref 53 is wrong in the table the N=21 not 59. this needs correction

Author Response

Point 1: The reference to Johahnnson 1992 ref 53 is wrong in the table the N=21 not 59. this needs correction

Response 1. Thank you for your valuable feedback. We corrected this mistake in the mentioned table.